# HiChunk: Evaluating and Enhancing Retrieval Augmented Generation with Hierarchical Chunking

## Abstract

Retrieval-Augmented Generation (RAG) enhances the response capabilities of language models by integrating external knowledge sources. However, document chunking as an important part of RAG system often lacks effective evaluation tools. This paper first analyzes why existing RAG evaluation benchmarks are inadequate for assessing document chunking quality, specifically due to evidence sparsity. Based on this conclusion, we propose HiCBench, which includes manually annotated multi-level document chunking points, synthesized evidence-dense question answer(QA) pairs, and their corresponding evidence sources. We also propose HiChunk, a hierarchical document structuring framework using fine-tuned LLMs and the Auto-Merge retrieval algorithm to enhance retrieval quality. Experiments demonstrate that HiCBench effectively evaluates the impact of different chunking methods across the entire RAG pipeline. Moreover, HiChunk achieves better chunking quality within reasonable time consumption, thereby enhancing the overall performance of RAG systems.

## 1 Introduction

RAG (Retrieval-Augmented Generation) enhances the quality of LLM responses to questions beyond their training corpus by flexibly integrating external knowledge through the retrieval of relevant content chunks as prompts(Lewis et al., 2020). This approach helps reduce hallucinations(Chen et al., 2024b; Zhang et al., 2025), especially when dealing with real-time information(He et al., 2022) and specialized domain knowledge(Wang et al., 2023; Li et al., 2023). Document chunking, a crucial component of RAG systems, significantly impacts the quality of retrieved knowledge and, consequently, the quality of responses. Poor chunking methods may separate continuous fragments, leading to information loss, or combine unrelated information, making it more challenging to retrieve relevant content. For instance, as noted in Bhat et al. (2025), the optimal chunk size varies significantly across different datasets.

Although numerous benchmarks exist for evaluating RAG systems(Bai et al., 2024; Dasigi et al., 2021; Duarte et al., 2024; Zhang et al., 2024; Yang et al., 2018b; Kočiský et al., 2018; Pang et al., 2021), they mostly focus on assessing either the retriever's capability or the reasoning ability of the response model, without effectively evaluating chunking methods. We analyzed several datasets to determine the average word and sentence count of evidence. As shown in Table 1, existing benchmarks generally suffer from evidence sparsity, where only a few sentences in the document are relevant to the query. As illustrated in Figure 1, this sparsity of evidence makes these datasets inadequate for evaluating the performance of chunking methods. In reality, user tasks might be evidence-dense, such as enumeration or summarization tasks, requiring chunking methods to accurately and completely segment semantically continuous fragments. Therefore, it is essential to effectively evaluate chunking methods.

To address this, we introduce **Hi**erarchical **C**hunking **B**enchmark(**HiCBench**), a benchmark for document QA designed to effectively evaluate the impact of chunking methods on different components of RAG systems, including the performance of document chunking, retrievers, and response models. HiCBench's original documents are sourced from OHRBench. We curated documents of appropriate length for the corpus and manually annotated chunking points at various hierarchical levels for

evaluation purposes. These points are used to assess the chunker's performance and construct QA pairs, followed by using LLMs and the annotated document structure to create evidence-dense QA, and finally extracting relevant evidence sentences and filtering non-compliant samples using LLMs.

Additionally, existing document chunking methods only consider linear document structure(Duarte et al., 2024; Xiao et al., 2024; Zhao et al., 2025; Wang et al., 2025), while user problems may involve fragments with different semantic granularity, and linear document structure makes it difficult to adaptively adjust during retrieval. Therefore, we propose the **Hi**erarchical **C**hunking framework(**HiChunk**), which employs fine-tuned LLMs for hierarchical document structuring and incorporates iterative reasoning to address the challenge of adapting to extremely long documents. For hierarchically structured documents, we introduce the Auto-Merge retrieval algorithm, which adaptively adjusts the granularity of retrieval chunks based on the query, thereby maximizing retrieval quality. In this work, our main contributions are as follows:

- We introduce HiCBench, a benchmark designed to assess the performance of chunker and the impact of chunking methods on retrievers and response models within RAG systems. HiCBench includes information on chunking points at different hierarchical levels of documents, as well as sources of evidence and factual answers related to evidence-dense QA, enabling better evaluation of chunking methods.

- We propose the HiChunk framework, a document hierarchical structuring framework that allows RAG systems to dynamically adjust the semantic granularity of retrieval chunks.

- We conduct comprehensive performance evaluations on several open-source datasets and HiCBench, analyzing the impact of different chunking methods across three dimensions: performance of chunker, retriever, and responder.

Table 1: Statistics of benchmarks.

| Dataset | Qasper | OHRBench | GutenQA |
|---------|--------|----------|---------|
| $\text{Num}_{doc}$ | 416 | 1261 | 100 |
| $\text{Sent}_d$ | 164 | 176 | 5,373 |
| $\text{Word}_d$ | 4.2k | 5.4k | 146.5k |
| $\text{Num}_{qa}$ | 1,372 | 8,498 | 3,000 |
| $\text{Word}_q$ | 8.9 | 20.6 | 16.0 |
| $\text{Word}_a$ | 16.0 | 5.6 | 26.0 |
| $\text{Word}_e$ | 239.4 | 36.5 | 39.3 |
| $\text{Sent}_e$ | 10.5 | 1.7 | 1.7 |

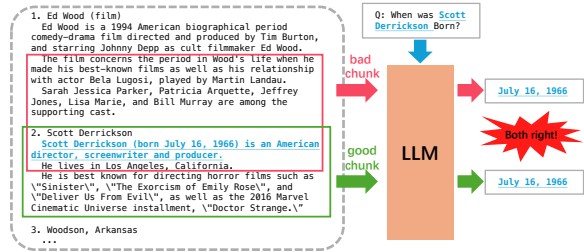

Figure 1: Different methods produce the same answer.

## 2 RELATED WORKS

**Traditional Text Chunking.** Text chunking divides continuous text into meaningful units like sentences, phrases, and words, with our focus on sentence-level chunking. Recent works have explored various approaches: (Cho et al., 2022) combines text chunking with extractive summarization using hierarchical representations and determinantal point processes (DPPs) to minimize redundancy, (Liu et al., 2021) presents a pipeline integrating topical chunking with hierarchical summarization, and (Zhang et al., 2021) develops an adaptive sliding-window model for ASR transcripts using phonetic embeddings. However, these LSTM and BERT(Devlin et al., 2019) based methods face limitations from small context windows and single-level chunking capabilities.

**RAG-oriented Document Chunking.** Recent research has explored content-aware document chunking strategies for RAG systems. LumberChunker(Duarte et al., 2024) uses LLMs to identify semantic shifts,but may miss hierarchical relationships. PIC(Wang et al., 2025) proposes pseudo-instruction for document chunking, guide chunking via document summaries, though its single-level approach may oversimplify document structure. AutoChunker(Jain et al., 2025) employs tree-based representations but primarily focuses on noise reduction rather than multi-level granularity. Late Chunking(Günther et al., 2024) embeds entire documents before chunking to preserve global context, but produces flat chunk lists without modeling hierarchical relationships. LongRefiner(Jin et al., 2025) introduced two-level chunking, but it is constrained by the model input length and hallucination issues. In contrast, our HiChunk method creates multi-level document representations, chunking from coarse sections to fine-grained paragraphs. This enables RAG systems to retrieve information at appropriate abstraction levels, effectively bridging fragmented knowledge gaps.

**Limitations of Existing Text Chunking Benchmarks.** The evaluation of text chunking and RAG methods heavily relies on benchmark datasets. Wiki-727k(Koshorek et al., 2018),VT-SSum(Lv et al., 2021) and NewsNet(Wu et al., 2023) are typically chunked into flat sequences of paragraphs or sentences, without capturing the multi-level organization (e.g., sections, subsections, paragraphs) inherent in many real-world documents. This single-level representation limits the ability to evaluate chunking methods that aim to preserve or leverage document hierarchy, which is crucial for comprehensive knowledge retrieval in complex RAG scenarios. While Qasper(Dasigi et al., 2021), HotpotQA(Yang et al., 2018a) and GutenQA(Duarte et al., 2024) are designed for RAG-related tasks, they do not specifically provide mechanisms or metrics for evaluating the efficacy of document chunking strategies themselves. Their focus is primarily on end-to-end RAG performance, where the impact of chunking is implicitly measured through retrieval and generation quality. This makes it challenging to isolate and assess the performance of different chunking methods independently, hindering systematic advancements in hierarchical document chunking. Our work addresses these gaps by proposing a method that explicitly considers multi-level document chunking and constructs a novel benchmark from a chunking perspective.

## 3 HiCBench Construction

In order to construct the HiCBench dataset, we performed additional document hierarchical structuring and created QA pairs to evaluate document chunking quality, building on the OHRBench document corpus(Zhang et al., 2024). It contains documents from various fields in the real world, such as academia, finance, law, manual, and so on. We filter documents with fewer than 4,000 words and those exceeding 50 pages. For retained documents, we manually annotated the hierarchical structure and used these annotations to assist in the generation of QA pairs and to assess the accuracy of document chunking.

**Task Criteria** To ensure that the constructed QA pairs could effectively evaluate the quality of document chunking, we aimed for the evidence associated with each QA pair to be widely distributed across a complete semantic chunk. Failure to fully recall such a semantic chunk would result in missing evidence, thereby degrading the quality of the generated responses. To achieve this objective, we established the following standards to regulate the generation of QA pairs:

- **Evidence Completeness and Density**: Evidence completeness ensures that the evidence relevant to the question is comprehensive and necessary within the context. Evidence density requires that evidence constitutes a significant proportion of the context, enhancing the QA pair's utility for evaluating chunking methods.
- **Fact Consistency**: To ensure the constructed samples can evaluate the entire retrieval-based pipeline, it is essential that the generated responses remain consistent with the answers when provided with full context, and that the questions are answerable.

**Task Definition** We define three different task types to evaluate the quality of chunking:

- **Evidence-Sparse QA** ($T_0$): The evidence related to the QA is confined to one or two sentences within the document.
- **Single-Chunk Evidence-Dense QA** ($T_1$): Evidence sentences related to the QA constitute a substantial portion of the context within a single complete semantic chunk. The chunk size ranges from 512 to 4096 tokens.
- **Multi-Chunk Evidence-Dense QA** ($T_2$): Evidence sentences related to the QA are distributed across multiple complete semantic chunks, covering a significant portion of the context. The chunk size ranges from 256 to 2048 tokens.

**QA Construction** We use a prompt-based approach using DeepSeek-R1-0528[1] to generate candidate QA pairs, followed by a series of filtering processes to ensure the retained QA pairs meet the criteria of evidence completeness, density, and fact consistency. The specific process is as follows:

---

[1]https://huggingface.co/deepseek-ai/DeepSeek-R1-0528

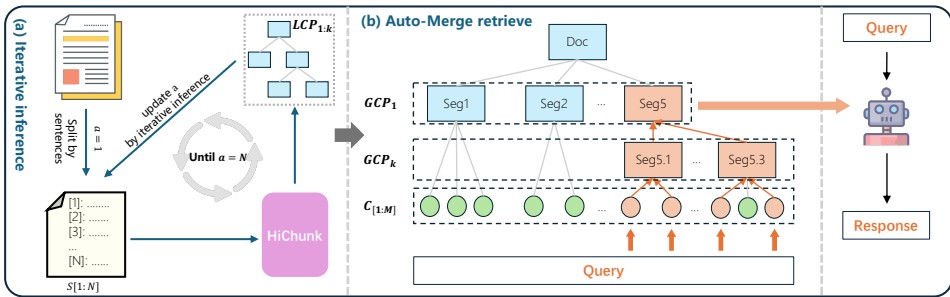

Figure 2: Overview of the proposed HiChunk framework.

1. **Document Hierarchical Annotation and Summarization**: To enable LLMs to gain an overall understanding of the specific document $D$ while constructing QA pairs, we first generated summaries for corresponding sections based on the annotated hierarchical structure, denoted as $S \leftarrow LLM_s(D)$. These summaries will be used in QA pair generation.

2. **Generation of Questions and Answers**: We randomly selected one or two chunks from all eligible document fragments as context $C$, then generated candidate QA pairs using $(S, C)$, where $(Q, A) \leftarrow LLM_{qa}(S, C)$.

3. **Ensuring Evidence Completeness and Density**: Referring to Friel et al. (2024), we use LLMs to extracted sentences from context $C$ related to the QA pair as evidence, denoted as $E \leftarrow LLM_{ee}(C, Q, A)$. To mitigate hallucination effects, this step will be repeated five times, retaining sentences that appeared at least four times as the final evidence. Furthermore, to ensure evidence density, we remove samples which the ratio of evidence is less than 10% of context $C$.

4. **Ensuring Fact Consistency**: We applied Fact-Cov metric(Xiang et al., 2025) to filter test samples. We first extract the facts from answer $A$, denoted as $F \leftarrow LLM_{fe}(Q, A)$[1]. Contexts $C$ used for constructing QA pairs will be provided to LLMs to generate response $R'$, denoted as $R' \leftarrow LLM_r(Q, C)$. Then, the Fact-Cov metric will be calculated by $\text{Fact\_Cov} \leftarrow LLM_{fc}(F, R')$[1]. This process will be repeated 5 times. We retain samples with an average Fact-Cov metric exceeding 80%. Samples below this threshold are deemed unanswerable. All prompts used for QA construction are provided in subsection A.6.

## 4 METHODOLOGY

This section primarily introduces the HiChunk framework. The overall framework is illustrated in Figure 2. The aim is for the fine-tuned LLMs to comprehend the hierarchical relationships within a document and ultimately organize the document into a hierarchical structure. This involves two subtasks: identification of chunking points and determination of hierarchy levels. Through prompts, HiChunk converts these two subtasks into text generation task. In model train of HiChunk, we use Gov-report(Huang et al., 2021), Qasper(Dasigi et al., 2021) and Wiki-727k(Koshorek et al., 2018) to construct training instructions, which are publicly available datasets with explicit document structure. Meanwhile, we augment the training set by randomly shuffling document chapters and deleting document content.

During inference, HiChunk first splits a document $D$ into a list of sentences $S = [s_1, s_2, ..., s_N]$ (each sentence is assigned a unique ID). The goal is to output a set of hierarchical chunk points that partition $S$ into non-overlapping, semantically complete chunks. Each chunk point is represented as a tuple: $(id, level)$, it represents a semantic break at a specific hierarchy level.

Although the chunking result of HiChunk has semantic integrity, the variability in the chunk length distribution caused by the semantic chunking method can lead to disparities in semantic granularity, which can affect retrieval quality. To mitigate this, we apply a fixed-size chunking approach on the results of HiChunk to produce $C_{[1:M]}$, and propose the Auto-Merge retrieval algorithm to balance issues of varying semantic granularity and the semantic integrity of retrieved chunks.

---

[1]https://github.com/GraphRAG-Bench/GraphRAG-Benchmark

**Iterative Inference** For documents exceeding the model's input length limit $L$, we employ a sliding window approach. In each iteration, we greedily select the longest possible text segment starting from the current position that fits within the limit $L$. The model then predicts local chunk points for this segment, which are subsequently aggregated into the global document structure.

However, iterative inference suffers from hierarchical drift phenomenon. Due to the lack of complete structural information about document, the model may incorrectly predict the first chunking point of the current inference process as a level-1 segment, thereby causing local hierarchical misalignment. To mitigate this problem, we construct residual text lines from known document structures to guide the model making correct hierarchical judgments. The complete iterative inference procedure is illustrated in algorithm 1.

**Auto-Merge Retrieval Algorithm** To balance the semantic richness and completeness of recalled contexts, we propose Auto-Merge retrieval algorithm. This algorithm uses a series of conditions to control the extent to which child nodes are merged upward into parent nodes. Auto-Merge algorithm traverses the query-ranked chunks $C_{[1:M]}^{sorted}$, using $\mathcal{N}$ to record the nodes that have been recalled. During the $i$-th step of the traversal, we first record the current used token budget, $T_{used} = \sum_{n \in \mathcal{N}} \text{len}(n)$. We then add $C_{[i]}^{sorted}$ to $\mathcal{N}$ and denote the parent of $C_{[i]}^{sorted}$ by $p$. Finally, we merge upward when the following conditions are met:

- **Coherence ($Cond_1$):** The retrieval set contains multiple children from the same parent. Formally, the number of retrieved children must be at least two: $|\mathcal{N} \cap \text{children}(p)| \geq 2$.
- **Substantiality ($Cond_2$):** The total length of the retrieved children covers a significant portion of the parent text. We require $\sum_{n \in (\mathcal{N} \cap \text{children}(p))} \text{len}(n) \geq \theta^* * \text{len}(p)$. Here, $\theta^*$ is an adaptive threshold defined as:

$$\theta^*(T_{used}, p) = \frac{1}{3} \times \left(1 + \frac{T_{used}}{T_{max}}\right)$$

where $T_{used}$ is the current token usage and $T_{max}$ is the total budget. This design ensures that $\theta^*$ starts low and increases as the budget fills up. Intuitively, this encourages **higher-ranking chunks** (processed when $T_{used}$ is low) to merge more aggressively, prioritizing structural integrity for the most relevant information.

- **Feasibility ($Cond_3$):** The remaining token budget is sufficient to accommodate the full parent node after replacing its children.

The detailed procedure is outlined in algorithm 2.

---

**Algorithm 1:** iterative inference

**input** : Document $D$, Input length $L$
**output:** Global chunk points $GCP_{1:k}$

1 $S[1 : N] \leftarrow \text{SentTokenize}(D)$;
2 $a \leftarrow 1$;
3 $b \leftarrow \text{argmax}_{\hat{b}}(S[a : \hat{b}] \leq L)$;
4 $res\_lines \leftarrow \text{None}$;
5 $GCP_{1:k} \leftarrow [] * k$ ;
6 **while** $1 \leq a < b \leq N$ **do**
7    $LCP_{1:k} \leftarrow \text{HiChunk}(S[a : b], res\_lines)$;
8    $GCP_{1:k} \leftarrow \text{Merge}(GCP_{1:k}, LCP_{1:k})$;
9    **if** $len(LCP_1) \geq 2$ **then**
10      $a \leftarrow LCP_1[-1]$;
11      $res\_lines \leftarrow \text{None}$;
12    **else**
13      $a \leftarrow b$;
14      $res\_lines \leftarrow \text{ResLines}(GCP_{1:k})$;
15    $b \leftarrow \text{argmax}_{\hat{b}}(S[a : \hat{b}] \leq L)$;
16 **return** $GCP_{1:k}$

---

**Algorithm 2:** retrieval algorithm

**input** : Token budget $T$, Chunks $C_{[1:M]}$, Query $q$
**output:** Retrieval context ctx

1 $C_{[1:M]}^{sorted} \leftarrow \text{Sorted}(C_{[1:M]}, q)$;
2 $\mathcal{N} \leftarrow [], T_{used} \leftarrow 0$;
3 **for** $i \leftarrow 1$ **to** $M$ **do**
4    $\mathcal{N} \leftarrow \mathcal{N} + C_{[i]}^{sorted}$;
5    ctx$, T_{used} \leftarrow \text{Context}(\mathcal{N})$;
6    $p \leftarrow \text{parent}(C_{[i]}^{sorted})$;
7    **while** $Cond_{[1,2,3]}$ **do**
8      **if** $T_{used} \geq T$ **then**
9        **break**
10      $\mathcal{N} \leftarrow \text{Merge}(\mathcal{N}, p)$;
11      ctx$, T_{used} \leftarrow \text{Context}(\mathcal{N})$;
12      $p \leftarrow \text{parent}(p)$;
13    **if** $T_{used} \geq T$ **then**
14      **break**
15 **return** ctx $[: T]$

## 5 EXPERIMENTS

### 5.1 DATASETS AND METRICS

The test subsets of Gov-report(Huang et al., 2021) and Qasper(Dasigi et al., 2021) datasets will be used for evaluation of chunking accuracy. For the Gov-report dataset, we only retain documents with document word count greater than 5k for experiments. To evaluate the accuracy of the chunking points, we use the $F1$ metrics of the chunking points. The $F1_{L_1}$ and $F1_{L_2}$ correspond to the chunking points of the level 1 and level 2 chunks, respectively. And the $F1_{L_{all}}$ metric does not consider the level of the chunking point. The Qasper, GutenQA(Duarte et al., 2024), and OHRBench(Zhang et al., 2024) datasets contain evidence relevant to the question. These datasets will be used in the evaluation for context retrieval.

For the full RAG pipeline evaluation, we used the publicly available datasets LongBench(Bai et al., 2024), Qasper, GutenQA, and OHRBench. the LongBench RAG evaluation contains 8 subsets from different datasets, with a total of 1,550 QA pairs, which can be categorized into single document QA and multiple document QA. The Qasper dataset contains 1,372 QA pairs from 416 documents. The GutenQA dataset contains 3,000 QA pairs based on 100 documents. In GutenQA, the average number of words in a document is 146,506, which is significantly higher than the other datasets. The documents of OHRBench come from seven different areas. We keep the documents with word counts greater than 4k in OHRBench and use the original QA pairs corresponding to these documents as a representative of the task $T_0$, denoted as OHRBench($T_0$). We use the F1 score and Rouge metrics to assess the quality of LLM responses. All experiments are conducted in the code repository of LongBench[2].

Furthermore, HiCBench will be used for comprehensive evaluation, including chunking accuracy, evidence recall rate, and RAG response quality assessment. To avoid biases from sparse text quality evaluation metrics, we employ the Fact-Cov(Xiang et al., 2025) metric for response quality evaluation of HiCBench. The Fact-Cov metric is repeatedly calculated 5 times to take the average. Statistics information of datasets used in experiment are shown in Table 2.

Table 2: Statistics of dataset used in experiments.

| Dataset | Qasper | GutenQA | OHRBench($T_0$) | HiCBench($T_1, T_2$) |
|---------|--------|---------|-----------------|----------------------|
| $\text{Num}_{doc}$ | 416 | 100 | 214 | 130 |
| $\text{Sent}_d$ | 164 | 5,373 | 886 | 298 |
| $\text{Word}_d$ | 4.2k | 146.5k | 26.8k | 8.5k |
| $\text{Num}_{qa}$ | 1,372 | 3,000 | 4,702 | (659, 541) |
| $\text{Word}_q$ | 8.9 | 16.0 | 22.2 | (31.0, 33.0) |
| $\text{Word}_a$ | 16.0 | 26.0 | 4.8 | (130.1, 126.4) |
| $\text{Word}_e$ | 239.4 | 39.3 | 39.1 | (561.5, 560.5) |
| $\text{Sent}_e$ | 10.5 | 1.7 | 1.7 | (20.5, 20.4) |

### 5.2 COMPARISON METHODS

We primarily compared two types of chunking methods: rule-based chunking methods and semantic-based chunking methods. All the comparison methods are as follows:

- **FC200**: Fixed chunking is a rule-based method, which first divide the document into sentences and then merge sentences based on a fixed chunking size. Here, the fixed chunking size is 200.

- **SC**: Semantic Chunker(Xiao et al., 2024) uses an embedding model to calculate the similarity between adjacent paragraphs for chunking. We use bge-large-en-v1.5(Xiao et al., 2024) as the embedding model.

- **LC**: LumberChunker(Duarte et al., 2024) employs LLMs to predict the positions for chunking. In our experiments, we use Deepseek-r1-0528(DeepSeek-AI, 2025) as the prediction model. The sampling temperature set to 0.1.

---

[2]https://github.com/THUDM/LongBench/tree/main

- **HC200**: HiChunk is the proposed method. In the model training for HiChunk. We further chunk the chunks of HiChunk by the fixed chunking method. The fixed chunking size is set to 200, denoted as HC200.
- **HC200+AM**: "+AM" represents the result of introducing Auto-Merge retrieval algorithm on the basis of HC200.

## 5.3 EXPERIMENTAL SETTINGS

In the model training of HiChunk, Gov-report(Huang et al., 2021), Qasper(Dasigi et al., 2021) and Wiki-727k(Koshorek et al., 2018) are the train datasets, which are publicly available datasets with explicit document structure. We use Qwen3-4B(Team, 2025) as the base model, with a learning rate of 1e-5 and a batch size of 64. The maximum length of training and inference is set to 8192 and 16384 tokens, respectively. Meanwhile, the length of each sentence is limited to within 100 characters. Due to the varying sizes of chunks resulting from semantic-based chunking, we limit the length of the retrieved context based on the number of tokens rather than the number of chunks for a fair comparison. The maximum length of the retrieved context is set to 4096 tokens. We also compare the performance of different chunking methods under different retrieved context length settings in subsection 5.6. In the RAG evaluation process, we consistently use Bge-m3(Chen et al., 2024a) as the embedding model for context retrieval. As for the response model, we use three different series of LLMs with varying scales: Llama3.1-8B(Dubey et al., 2024), Qwen3-8B, and Qwen3-32B(Team, 2025).

## 5.4 CHUNKING ACCURACY

To comprehensively evaluate the performance of the semantic-based chunking method, we conducted experiments using two publicly available datasets, along with the proposed benchmark, to assess the cut-point accuracy of the chunking method. Since the SC and LC chunking methods are limited to performing single-level chunking, we evaluated only the F1 scores for the initial level of chunking points and the F1 scores without regard for the hierarchy of chunking points. The evaluation results are presented in Table 3. In the Qasper and Gov-report datasets, which serve as in-domain test sets, the HC method shows a significant improvement in chunk accuracy compared to the SC and LC methods. Additionally, in HiCBench, an out-of-domain test set, the HC method exhibits even more substantial accuracy improvements. These findings demonstrate that HC enhances the base model's performance in document chunking by focusing exclusively on the chunking task. Moreover, as indicated in the subsequent experimental results presented in subsection 5.5, the accuracy improvement of the HC method in document chunking leads to enhanced performance throughout the RAG pipeline. This includes improvements in the quality of evidence retrieval and model responses.

Table 3: Chunking accuracy. **HC** means the result of HiChunk without fixed-size chunking. The best result is in **bold**.

| Chunk Method | Qasper | | | Gov-Report | | | HiCBench | | |
|---|---|---|---|---|---|---|---|---|---|
| | $F1_{L_1}$ | $F1_{L_2}$ | $F1_{L_{all}}$ | $F1_{L_1}$ | $F1_{L_2}$ | $F1_{L_{all}}$ | $F1_{L_1}$ | $F1_{L_2}$ | $F1_{L_{all}}$ |
| SC | 0.0759 | - | 0.1007 | 0.0298 | - | 0.0616 | 0.0487 | - | 0.1507 |
| LC | 0.5481 | - | 0.6657 | 0.1795 | - | 0.5631 | 0.2849 | - | 0.4858 |
| HC | **0.6742** | **0.5169** | **0.9441** | **0.9505** | **0.8895** | **0.9882** | **0.4841** | **0.3140** | **0.5450** |

## 5.5 RAG-PIPELINE EVALUATION

We evaluated the performance of various chunking methods on the LongBench, Qasper, GutenQA, OHRBench, and HiCBench datasets, with the results detailed in Table 4. The performance of each subset in LongBench is shown in Table A1. The results demonstrate that the HC200+AM method achieves either optimal or suboptimal performance on most LongBench subsets. When considering average scores, LumberChunk remains a strong baseline. However, as noted in Table 2, both GutenQA and OHRBench datasets exhibit the feature of evidence sparsity, meaning that the evidence related to QA pairs is derived from only a few sentences within the document. Consequently, the different chunking methods show minimal variation in evidence recall and response quality metrics on these datasets. For instance, using Qwen3-32B as the response model on the GutenQA

dataset, the evidence recall metrics of FC200 and HC200+AM are 64.5 and 65.53, and the Rouge metrics are 44.86 and 44.94, respectively. Another example is OHRBench dataset, the evidence recall metrics and Rouge metrics of FC200, LC, HC200 and HC200+AM are very close. In contrast, the Qasper and HiCBench datasets contain denser evidence, where a better chunking method results in higher evidence recall and improved response quality. Again using Qwen3-32B as an example, on the $T_1$ task of HiCBench dataset, the evidence recall metric for FC200 and HC200+AM are 74.06 and 81.03, the Fact-Cov metrics are 63.20 and 68.12, and the Rouge metrics are 35.70 and 37.29, respectively. These findings suggest that the evidence-dense QA in the HiCBench dataset is better suited for evaluating the quality of chunking methods, enabling researchers to more effectively identify bottlenecks within the overall RAG pipeline.

Table 4: RAG-pipeline evaluation results (ERec: Evidence Recall, FC: Fact Coverage). The best result is in **bold**, and the sub-optimal result is in underlined

| Chunk Method | LongBench Score | Qasper | | GutenQA | | OHRBench($T_0$) | | HiCBench($T_1$) | | | HiCBench($T_2$) | | |
|---|---|---|---|---|---|---|---|---|---|---|---|---|---|
| | | ERec | F1 | ERec | Rouge | ERec | Rouge | ERec | FC | Rouge | ERec | FC | Rouge |
| Llama3.1-8B | | | | | | | | | | | | | |
| FC200 | 42.49 | 84.08 | 47.26 | 64.43 | 30.03 | 67.03 | 51.01 | 74.84 | 47.82 | 28.43 | 74.61 | 46.79 | 30.97 |
| SC | 42.12 | 82.08 | 47.47 | 58.30 | 28.58 | 62.65 | 49.10 | 72.14 | 46.80 | 28.43 | 73.49 | 45.28 | 30.92 |
| LC | 42.73 | 87.08 | 48.20 | 63.67 | 30.22 | **68.42** | 51.85 | 76.64 | 50.84 | 29.62 | 76.12 | 49.12 | 32.01 |
| HC200 | **43.17** | 86.16 | 48.09 | 65.13 | 29.95 | 68.25 | 51.33 | 78.52 | 49.87 | 29.38 | 78.76 | 49.11 | 31.80 |
| +AM | 42.90 | 87.49 | 48.95 | 65.47 | 30.33 | 67.84 | 51.92 | 81.59 | 55.58 | 30.04 | 80.96 | 53.66 | 33.04 |
| Qwen3-8B | | | | | | | | | | | | | |
| FC200 | 43.95 | 84.32 | 45.10 | 64.50 | 33.47 | 67.07 | 48.18 | 74.06 | 47.35 | 33.83 | 72.95 | 43.45 | 35.27 |
| SC | 43.54 | 82.22 | 44.55 | 58.37 | 32.71 | 62.18 | 46.79 | 71.42 | 46.07 | 33.30 | 72.36 | 42.97 | 34.76 |
| LC | **44.83** | 87.43 | **46.05** | 63.67 | 33.87 | **68.79** | 49.28 | 75.53 | 48.27 | 34.12 | 75.14 | 46.80 | 35.93 |
| HC200 | 43.90 | 86.49 | 45.95 | 65.20 | 33.89 | 68.57 | 49.06 | 77.68 | 47.37 | 34.30 | 78.10 | 46.20 | 36.32 |
| +AM | 44.41 | 87.85 | 45.82 | 65.53 | 34.15 | 68.31 | 49.61 | 81.03 | 50.75 | 35.26 | 80.65 | 49.02 | 37.28 |
| Qwen3-32B | | | | | | | | | | | | | |
| FC200 | 46.33 | 84.32 | 46.49 | 64.50 | 44.86 | 67.07 | 46.89 | 74.06 | 63.20 | 35.70 | 72.95 | 60.87 | 37.17 |
| SC | 46.29 | 82.22 | 46.39 | 58.37 | 43.59 | 62.18 | 45.43 | 71.26 | 61.09 | 35.64 | 72.36 | 59.23 | 37.09 |
| LC | **47.43** | 87.43 | 46.82 | 63.67 | 44.45 | **68.79** | **47.92** | 75.53 | 64.76 | 36.15 | 75.14 | 62.75 | 38.02 |
| HC200 | 46.71 | 86.49 | 46.99 | 65.20 | 44.83 | 68.57 | 47.71 | 77.68 | 63.93 | 36.55 | 78.10 | 62.51 | 38.26 |
| +AM | 46.92 | 87.85 | 47.25 | 65.53 | 44.94 | 68.31 | 47.89 | 81.03 | 68.12 | 37.29 | 80.65 | 66.36 | 39.37 |

## 5.6 INFLUENCE OF RETRIEVAL TOKEN BUDGET

Since HiCBench is more effective in assessing the performance of chunking methods, we evaluated the impact of our proposed method on the $T_1$ task of HiCBench under different retrieve token budgets: 2k, 2.5k, 3k, 3.5k and 4k tokens. We compared the effects of various chunking methods by calculating the Rouge metrics between responses and answers, as well as the Fact-Cov metrics. The experimental findings are illustrated in Figure 3. The results demonstrate that a larger retrieval token budget usually leads to better response quality, so it is necessary to compare different chunking methods under the same retrieval token budget. HC200+AM consistently achieves superior response quality across various retrieve token budget settings. These experimental results underscore the effectiveness of HC200+AM method. We further present the correspond curves of the evidence recall metrics in subsection A.2.

## 5.7 EFFECT OF MAXIMUM HIERARCHICAL LEVEL

In this section, we examine the impact of limiting the maximum hierarchical level of document structure obtained by HiChunk. The maximum level ranges from 1 to 4, denoted as $L1$ to $L4$, while $LA$ represents no limitation on the maximum level. We measure the evidence recall metric on different settings. As shown in Figure 4. This result reveals that the Auto-Merge retrieval algorithm degrades the performance of RAG system in the $L1$ setting due to the overly coarse-grained semantics of $L1$ chunks. As the maximum level increases from 1 to 3, the evidence recall metric also gradually improves and remains largely unchanged thereafter. These findings highlight the importance of document hierarchical structure for enhancing RAG systems.

## 5.8 TIME COST FOR CHUNKING

As document chunking is essential for RAG systems, it must meet specific timeliness requirements. In this section, we analyze the time costs associated with different semantic-based chunking meth-

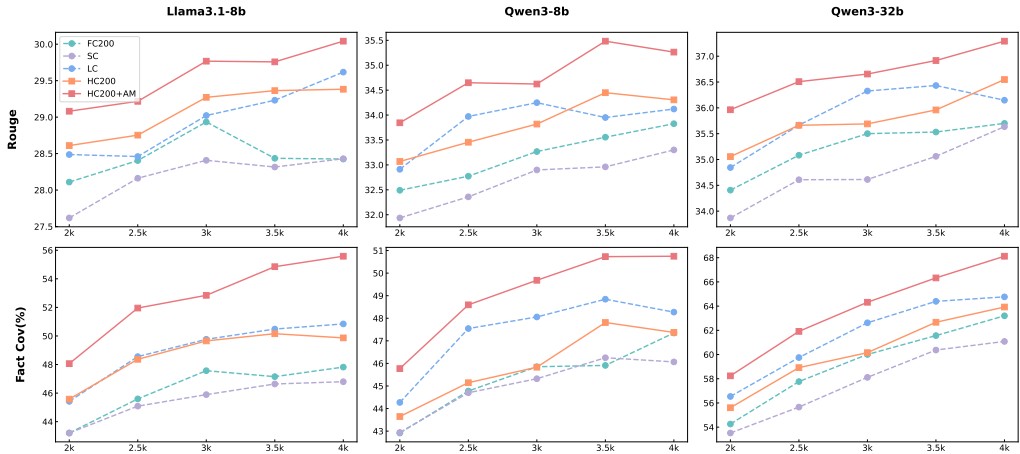

Figure 3: Performance of HiCBench($T_1$) under different retrieval token budget from 2k to 4k.

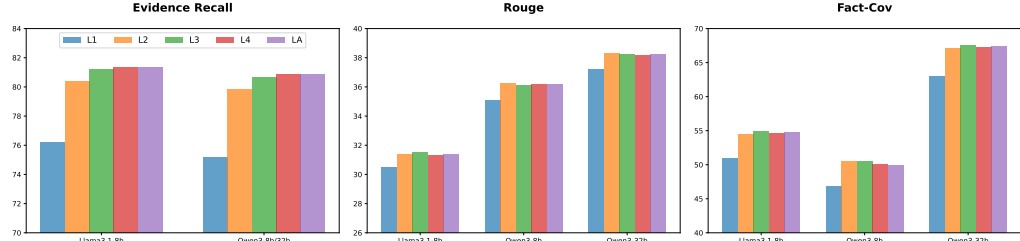

Figure 4: Evidence recall metric across different maximum level on HiCBench($T_1$ and $T_2$).

ods, as presented in Table 5. Although the SC method exhibits superior real-time performance, it consistently falls short in quality across various datasets compared to other baselines. However, the LC method demonstrates reasonably good performance, but its chunking speed is considerably slower than other semantic-based methods, limiting its applicability within RAG systems. In contrast, the HC method achieves the highest chunking quality among all baseline methods while maintaining an acceptable time cost, making it well-suited for implementation in real scenarios.

Table 5: Time cost of different chunking methods.

| Dataset | Avg. Word | SC | | LC | | HC | |
|---|---|---|---|---|---|---|---|
| | | Time(s/doc) | Chunks | Time(s/doc) | Chunks | Time(s/doc) | Chunks |
| Qasper | 4,166 | 0.4867 | 43.83 | 5.4991 | 18.32 | 1.4993 | 15.08 |
| Gov-report | 13,153 | 1.3219 | 114.72 | 15.4321 | 40.89 | 4.3382 | 29.79 |
| OHRBench($T_0$) | 26,808 | 3.0943 | 249.14 | 37.3935 | 89.68 | 14.5776 | 92.23 |
| GutenQA | 146,507 | 16.5028 | 1,453.00 | 132.4900 | 393.52 | 60.1921 | 232.85 |
| HiCBench | 8,519 | 1.0169 | 80.12 | 13.4414 | 41.48 | 5.7506 | 51.35 |

## 5.9 ABLATION STUDY FOR AUTO-MERGE

To verify the necessity and robustness of the rule design in the Auto-Merge algorithm, we conducted ablation experiments on its core merging conditions, using Qwen3-8B as the generator. The results are presented in Table 6.

When only $Cond_3$ (token budget constraint) is retained, the algorithm achieves optimal performance on evidence-dense tasks (HiCBench), with ERec of 81.43, Rouge of 36.33, and Fact-Cov of 51.35. However, its performance degrades notably on evidence-sparse tasks: the LongBench Score drops to 43.25, and the ERec on OHRBench is only 66.72. This indicates that relying solely on a single rule leads to poor generalization across diverse task types, lacking sufficient robustness.

Table 6: Ablation study for merging conditions of Auto-Merge.

| Condition Combination | HiCBench($T_1$ and $T_2$) | | | LongBench | Qasper | | OHRBench($T_0$) | |
|---|---|---|---|---|---|---|---|---|
| | ERec | Rouge | Fact-Cov | Score | ERec | F1 | ERec | Rouge |
| $Cond_3$ Only | 81.43 | 36.33 | 51.35 | 43.25 | 86.73 | 45.29 | 66.72 | 48.78 |
| $Cond_3 + Cond_1$ | 80.55 | 36.08 | 50.70 | 43.80 | 87.54 | 45.83 | 68.18 | 49.56 |
| $Cond_3 + Cond_1 + Cond_2$ | 80.86 | 36.17 | 49.97 | 44.41 | 87.85 | 45.82 | 68.31 | 49.61 |

After adding $Cond_1$ (semantic intersection constraint), the ERec of Qasper and OHRBench increases by 0.81 and 1.46, respectively, proving that semantic intersection constraints can mitigate "meaningless merging", thereby enhancing retrieval accuracy for evidence-sparse tasks.

With the addition of $Cond_2$ (length ratio constraint), the performance across all datasets tends to be balanced: LongBench Score increases to 44.41 (increases by 1.16), while HiCBench performance only slightly decreases (ERec decrease by 0.57). These results confirm that the combination of multiple complementary rules enables the Auto-Merge algorithm to adapt to both evidence-dense and evidence-sparse tasks, significantly improving its robustness. Furthermore, we conducted a sensitivity analysis on the threshold $\theta^*$ of $Cond_2$, and the detailed results are provided in subsection A.3.

## 5.10 COMBINATION WITH LATE-CHUNKING

In order to verify the complementarity of HiChunk with other optimization techniques, we supplemented the combination of Late-Chunking with various chunking methods and conducted experiments on HiCBench. The experiment setting is consistent with Günther et al. (2024), using jina-embeddings-v3(Sturua et al., 2024) as the embedding model. The results are presented in Table 7.

Table 7: The performance of combining the Late-Chunking and different chunking methods on HiCBench($T_1$ and $T_2$). The best result is in **bold**, and the sub-optimal result is in underlined

| Methods | w/o Late-Chunking | | | w/ Late-Chunking | | |
|---|---|---|---|---|---|---|
| | ERec | Rouge | Fact-Cov | ERec | Rouge | Fact-Cov |
| C200 | 75.59 | 34.19 | 46.71 | 78.04 | 34.33 | 49.12 |
| SC | 73.07 | 34.17 | 45.60 | 78.07 | 34.16 | 48.45 |
| LC | 77.89 | 34.84 | 49.16 | 79.93 | 35.16 | 50.65 |
| HC200 | 78.13 | 34.93 | 48.03 | 79.29 | 34.84 | 49.77 |
| HC200+AM | **80.87** | **36.34** | **51.49** | **81.20** | **36.00** | **52.71** |

Late-Chunking universally enhances the ERec and Fact-Cov metrics of various chunking methods. Regardless of whether Late-Chunking is integrated, HC200+AM consistently delivers the best performance across all evaluated settings. This result validates the flexibility of the HiChunk framework, whose design enables seamless integration with other RAG optimization techniques (e.g., Late-Chunking) to further boost end-to-end performance.

## 6 CONCLUSION

This paper begins by analyzing the shortcomings of current benchmarks used for evaluating RAG systems, specifically highlighting how evidence sparsity makes them unsuitable for assessing different chunking methods. As a solution, we introduce HiCBench, a QA benchmark focused on hierarchical document chunking, which effectively evaluates the impact of various chunking methods on the entire RAG process. Additionally, we propose the HiChunk framework, which, when combined with the Auto-Merge retrieval algorithm, significantly enhances the quality of chunking, retrieval, and model responses compared to other baselines.

## 7 REPRODUCIBILITY STATEMENT

To ensure the reproducibility of this work, we provide the complete data, code, and environment required for the experiment, as well as detailed descriptions of the entire experimental process in `https://anonymous.4open.science/r/HiChunk`:

- The complete datasets used in experiment, including proposed HiCBench dataset, is available on `./dataset` directory.
- The complete code for model training, inference, the Auto-Merge retrieval algorithm, and evaluation pipelines is available on `./pipeline` directory.

By providing the aforementioned resources and details, we aim to empower the research community to fully reproduce our results, build upon our work, and advance the field of retrieval-augmented generation. All materials are carefully anonymous under the double-blind review process to maintain the integrity of the review.

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

# A APPENDIX

## A.1 DETAIL OF LONGBENCH

In this section, we present the metric of each subset of the different chunking methods on Long-Bench, and the results are shown in Table A1.

Table A1: RAG-pipeline evaluation on LongBench and each subset. The best result is in **bold**, and the sub-optimal result is in underlined. Qasper* is the subset of LongBench.

| Chunk Method | Single-Doc QA | | | | Multi-Doc QA | | | | Avg |
|---|---|---|---|---|---|---|---|---|---|
| | NarrativeQA | Qasper* | MFQA-en | MFQA-zh | HotpotQA | 2WikiM | MuSiQue | DuReader | |
| Llama3.1-8B | | | | | | | | | |
| FC200 | **24.59** | 42.68 | 52.54 | 56.14 | 56.81 | 46.66 | 29.99 | 30.51 | 42.49 |
| SC | **24.59** | 42.12 | 52.10 | 57.43 | 54.34 | 45.44 | 30.24 | 30.68 | 42.12 |
| LC | 22.93 | 42.64 | 52.65 | 58.54 | 55.85 | 47.00 | 31.58 | 30.68 | 42.73 |
| HC200 | 23.75 | 43.57 | **54.04** | 57.51 | 56.52 | **48.29** | 31.06 | 30.65 | **43.17** |
| +AM | 24.46 | **43.85** | 52.10 | 56.65 | **57.27** | 46.24 | **31.82** | **30.84** | 42.90 |
| Qwen3-8B | | | | | | | | | |
| FC200 | 22.60 | 44.47 | 53.46 | 57.26 | 61.13 | 48.63 | 36.59 | 27.43 | 43.95 |
| SC | 24.73 | 43.69 | 52.83 | 58.66 | 56.22 | 46.77 | 37.83 | 27.59 | 43.54 |
| LC | 24.55 | 43.41 | **54.58** | 59.60 | 60.50 | **51.00** | 37.37 | 27.60 | **44.83** |
| HC200 | 21.96 | 42.38 | 51.23 | 58.47 | **62.84** | 49.57 | 38.03 | 26.74 | 43.90 |
| +AM | **21.79** | **46.37** | 52.81 | 58.86 | 61.94 | 47.17 | **39.09** | 27.28 | 44.41 |
| Qwen3-32B | | | | | | | | | |
| FC200 | 26.09 | 43.70 | **50.87** | 60.44 | **63.61** | 58.03 | 39.40 | 28.50 | 46.33 |
| SC | 26.19 | 43.47 | 49.54 | 61.63 | 61.37 | 58.13 | 40.65 | **29.34** | 46.29 |
| LC | 26.35 | **44.75** | 50.21 | **63.01** | 63.31 | 60.22 | **42.69** | 28.91 | **47.43** |
| HC200 | **27.01** | 44.44 | 49.69 | 62.16 | 61.85 | 61.24 | 38.54 | 28.54 | 46.71 |
| +AM | 26.97 | 44.49 | 50.28 | 60.47 | 61.67 | **62.37** | 40.80 | 28.29 | 46.92 |

## A.2 EVIDENCE RECALL UNDER DIFFERENT TOKEN BUDGET

In this section, we further present the curve of evidence recall metric at different retrieval context length settings (from 2k to 4k). The results are shown in Figure A1. Compared with other chunking methods, the HC200+AM method always maintains the best performance.

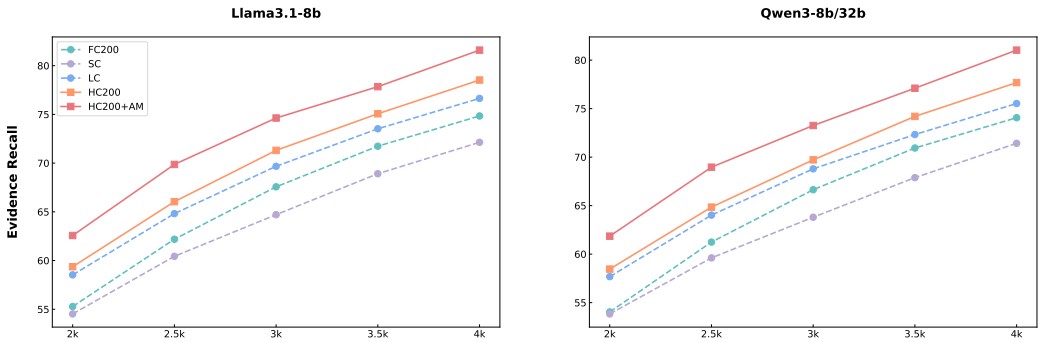

Figure A1: Evidence recall metric across different token budget on HiCBench($T_1$).

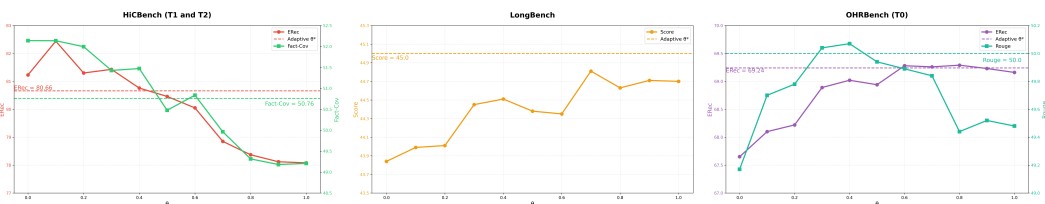

Figure A2: Performance changes across different $\theta$ on HiCBench, LongBench, and OHRBench. Dashed line represents the result of the adaptive $\theta^*$.

### A.3 SENSITIVITY ANALYSIS ON THRESHOLD $\theta^*$ OF $Cond_2$

The physical meaning of $\theta^*$ in $Cond_2$ is the minimum ratio of the total length of child nodes to the parent node length (controlling merging granularity). In order to verify the robustness of the Auto-Merge algorithm. We conduct the experiment by fixing $\theta$ from 0.0 to 1.0 (with an interval of 0.1) to test performance changes on three datasets. We use Qwen3-8B as generator model. The results are as Figure A2.

When $\theta$ ranges from 0.1 to 0.6, HiCBench's ERec remains above 80.05% and OHRBench's Rouge remains above 49.89%, indicating that the algorithm is robust to threshold variations; As $\theta$ increases, the performance of evidence-dense tasks decreases (ERec drops to 78.08% at $\theta$=1.0), while the performance of evidence-sparse tasks improves when $\theta > 0.5$, reflecting the granularity demand differences between the two types of tasks; The adaptive threshold $\theta^*$ achieves cross-task balance through dynamic adjustment: it maintains high evidence recall on HiCBench (80.66%) while achieving the optimal Score (45.00) on LongBench and Rouge (50.00) on OHRBench. This proves that it can adapt to different tasks without manual parameter tuning, with better robustness than fixed thresholds.

### A.4 FEW-SHOT PROMPTING EXPERIMENTS

To verify the necessity of fine-tuning, we have supplemented few-shot prompted experiments on HiCBench dataset. The base model is Qwen3-4B (consistent with the base model in the paper). We set two scenarios (1-shot and 3-shot) to compare their performance with the fine-tuned HiChunk, thereby validating the core value of fine-tuning for hierarchical chunking tasks. We use Qwen3-8B to generate response. The supplementary experimental results are presented in Table A2.

Table A2: Comparison of Chunking Accuracy and End-to-End RAG Performance: Few-Shot Prompting (1-shot/3-shot) vs. Fine-Tuned HiChunk on HiCBench.

| Method | Chunking Accuracy | | | RAG Performance | | | | | |
|---|---|---|---|---|---|---|---|---|---|
| | $F1_{L1}$ | $F1_{L2}$ | $F1_{ALL}$ | w/o Auto-Merge | | | w/ Auto-Merge | | |
| | | | | ERec | Rouge | Fact-Cov | ERec | Rouge | Fact-Cov |
| $HC_{1-shot}$ | 0.1784 | 0.1128 | 0.2328 | 72.35 | 33.14 | 44.02 | 73.47 | 34.53 | 46.62 |
| $HC_{3-shot}$ | 0.2500 | 0.1203 | 0.2199 | 72.26 | 33.08 | 43.97 | 73.05 | 34.04 | 46.55 |
| $HC_{ft}$ | **0.4841** | **0.3140** | **0.5450** | **77.87** | **35.21** | **46.84** | **80.86** | **36.17** | **49.97** |

The experimental results demonstrate that increasing the number of few-shot examples did not effectively improve the model's performance in chunking accuracy or the full-link performance of the subsequent RAG pipeline. Furthermore, all few-shot schemes show a significant performance gap compared to the fine-tuned HiChunk method. This fully confirms that fine-tuning is a necessary prerequisite for achieving high-quality hierarchical chunking of HiChunk.

### A.5 USE OF LLMS IN WRITING

In paper writing, AI tools are used for the following purposes: (1) Grammar checking and identifying word inconsistencies. (2) Polishing writing to improve fluency of the paper. Notably, the conception, development, and finalization of this research are completed entirely by the authors. AI tools were utilized solely for auxiliary purposes, and under no circumstances were they involved in

core scientific reasoning or decision-making. The authors have meticulously reviewed and edited all content to ensure its validity and alignment with their original intent, thereby guaranteeing the academic integrity of this work.

## A.6 PROMPTS

Listing A1: Prompt for segment summarization.

```
**Task:**
You are tasked with analyzing the provided document sections and their
    hierarchical structure. Your goal is to generate a concise and
    informative paragraph describing the content of each section and
    subsection.

**Instructions:**
1. Each section or subsection is identified by a header in the format
    `===SECTION xxx===` (for example, `===SECTION 1===`, `===SECTION
    2.1===`, etc.).
2. For every section and subsection, write a brief, clear, and
    informative paragraph summarizing its content. Do not omit any
    section or subsection.
3. Present your output as a JSON object with the following structure:
```json
{
    "SECTION 1": "description of section 1",
    "SECTION 1.1": "description of section 1.1",
    ...
    "SECTION n.m": "description of section n.m"
}
```
4. Ensure that each key in the JSON object matches the exact section
    identifier (e.g., `"SECTION 2.1.3"`), and  do not include any
    sections or subsections that are not present in the provided
    document fragment.
5. Do not add any commentary or explanation outside the JSON object.

**Document Fragment:**
```

Listing A2: Prompt for QA cunstruction.

```
You are provided with a document that includes a detailed structure of
    sections and subsections, along with descriptions for each.
    Additionally, complete contents are provided for a few selected
    sections. Your task is to create a question and answer pair that
    effectively captures the essence of the selected sections. Finally,
    you need to extract the facts which are mentioned in the answer.

<Type of Generated Q&A Task: Evidence-dense Dependent Understanding task>
Understanding task means that, the generated question-answering pairs
    that require the responser to extract information from documents.
    The answer should be able to find directly in the documents without
    any reasoning.
Evidence-dense dependent means that the facts about generated question
    are wildly distributed across all parts of the retrieved sections.

<Criteria>
- The question MUST be detailed and be based explicitly on information
    in the document.
- The question MUST include at least one entity.
- Question must not contain any ambiguous references, such as 'he',
    'she', 'it', 'the report', 'the paper', and 'the document'. You MUST
    use their complete names.
- The context sentence the question is based on MUST include the name of
    the entity. For example, an unacceptable context is "He won a bronze
```

```
medal in the 4 * 100 m relay". An acceptable context is "Nils
Sandstrom was a Swedish sprinter who competed at the 1920 Summer
Olympics."
- **THE MOST IMPORTANT: Evidence-dense dependency**, Questions must
require understanding of ENTIRE selected sections. Never base Q&A on
isolated few sentences. For example, a question comply the
**Evidence-dense dependency** criteria means that the facts about
this question should be wildly distributed across all parts of the
retrieved sections.

<Output Format>
Your response should be structured as follows:
```json
{{
    "question": "Your generated question here",
    "answer": "Your generated answer here"
}}
```

<Document Structure and Description>
{section_description}

<Retrieved Section and Content>
{section_content}
```

Listing A3: Prompt for evidence retrieval.

```
**Task:**
Analyze the relationship between context sentences and answer sentences.

**Instructions:**
1. You are given:
    - A context fragment, with each sentence numbered as follows:
    `[serial number]: context sentence content`
    - A question and its corresponding answer, with each answer sentence
    numbered as follows: `<serial number>: answer sentence content`
2. For each sentence in the answer, identify which sentence(s) from the
    context provide the information used to construct that answer
    sentence.
3. Present your findings in the following JSON format:
```json
{{
    "<answer_sentence_id_1>": "[context_sentence_id_1], ...,
    [context_sentence_id_n]",
    "<answer_sentence_id_2>": "[context_sentence_id_1], ...,
    [context_sentence_id_m]",
    ...
    "<answer_sentence_id_i>": "[context_sentence_id_1], ...,
    [context_sentence_id_j]"
}}
```

**Notes:**
- Only include answer sentences that have supporting evidence in the
    context.
- If an answer sentence does not have a source in the context, do not
    include it in the JSON output.
- Use only the serial numbers (not the full sentences) for both context
    and answer sentences in your JSON output.
- If multiple context sentences support an answer sentence, list all
    relevant context sentence numbers, separated by commas.

**Context Sentences:**
{context_sentence_list}
```

```
**Question:**
{question}

**Answer Sentences:**
{answer_sentence_list}
```

Listing A4: Prompt for model training.

```
You are an assistant good at reading and formatting documents, and you
    are also skilled at distinguishing the semantic and logical
    relationships of sentences between document context. The following
    is a text that has already been divided into sentences. Each line is
    formatted as: "{line number} @ {sentence content}". You need to
    segment this text based on semantics and format. There are multiple
    levels of granularity for segmentation, the higher level number
    means the finer granularity of the segmentation. Please ensure that
    each Level One segment is semantically complete after segmentation.
    A Level One segment may contain multiple Level Two segments, and so
    on. Please incrementally output the starting line numbers of each
    level of segments, and determine the level of the segment, as well
    as whether the content of the sentence at the starting line number
    can be used as the title of the segment. Finally, output a list
    format result, where each element is in the format of: "{line
    number}, {segment level}, {be a title?}".

>>> Input text:
```

