# OpenReview forum: "HiChunk: Evaluating and Enhancing Retrieval-Augmented Generation with Hierarchical Chunking"
_ICLR.cc/2026/Conference — ICLR 2026 Conference Withdrawn Submission_

### Official Review · Reviewer_Sj9B · 2025-10-29

**Soundness:** 2
**Presentation:** 3
**Contribution:** 1
**Rating:** 4
**Confidence:** 4

**Summary:**

This study proposes the HiChunk framework, which employs a fine-tuned LLM to parse the hierarchical structure of documents, thereby overcoming the limitations of traditional linear chunking. The Auto-Merge retrieval algorithm is introduced, which can dynamically adjust the retrieval granularity based on queries and adaptively merge hierarchical chunks. The paper validates the necessity of evidence-intensive evaluation through HiCBench and demonstrates the advantages of the combination of HiChunk and Auto-Merge.

**Strengths:**

1. The paper identifies and defines the evidence sparsity issue in existing RAG benchmarks. When the answers to QA pairs rely solely on a few sporadic sentences in the document, existing benchmarks fail to effectively evaluate the performance of chunking strategies.
2. The experiments in Table 4 demonstrate the effectiveness of HiCBench. On traditional sparse benchmarks, the performance differences among various chunking methods are negligible, whereas on HiCBench, the advantages of HC200+AM are amplified.
3. The paper proves the superiority of HiChunk in multiple dimensions, including chunking accuracy, evidence recall rate, and RAG response quality.

**Weaknesses:**

1. Studies such as LongRefiner[1] have already proposed leveraging the hierarchical nature of documents to optimize RAG. The HiChunk framework presented in this paper shares similarities with such hierarchical document optimization approaches.
2. The merging decisions of the Auto-Merge algorithm rely on a series of sophisticated rules. However, the paper does not provide ablation experiments or sensitivity analyses for these critical design elements. This renders the algorithm akin to a product fine-tuned for specific tasks, and its robustness remains to be further verified.
3. Furthermore, the paper does not explore whether a LLM can achieve equally proficient hierarchical structure parsing solely through prompting. It is likely that the fine-tuned model is merely reproducing capabilities it already possesses, thus casting doubt on the necessity of fine-tuning.

**reference**

[1] Jin, Jiajie, et al. "Hierarchical document refinement for long-context retrieval-augmented generation." arXiv preprint arXiv:2505.10413 (2025).

**Questions:**

1. If HiChunk has already generated semantically complete text chunks, why not directly perform retrieval on these finest-grained semantic chunks?
2. Many real-world documents possess a depth structure far exceeding three levels. How should this situation be addressed?
3. During the training of the HiChunk model, has it failed to learn effective modeling and differentiation of deeper hierarchical structures beyond L3?

---

> ### Author Response · Authors · 2025-11-21
>
> We sincerely thank the reviewers valuable and insightful comments. Our responses are as follows:
>
> ***
>
> **W1:** Studies such as LongRefiner have already proposed leveraging the hierarchical nature of documents to optimize RAG. The HiChunk framework presented in this paper shares similarities with such hierarchical document optimization approaches.
>
> **A1:** There are core differences between LongRefiner and HiChunk, highlighting the unique design and advantages of HiChunk:
>
> 1. Difference in long-document adaptation capability: LongRefiner requires the complete document as input during inference and then outputs a document with structural markers via a fine-tuned LLM. This limits it to the model’s maximum input length, making it unable to handle extremely long documents. In contrast, HiChunk adopts an iterative inference strategy, controlling the input length of each model run within a reasonable range through block-wise processing, enabling flexible adaptation to hierarchical chunking tasks for long documents.
> 2. LongRefiner requires structured output of the original document. Although it omits irrelevant content, its total output tokens are far more than those of HiChunk, and it exhibits a higher hallucination rate. Through modeling transformation, we first perform sentence segmentation on the input document, where each sentence is assigned a unique ID. This converts the chunking task into a classification task—we only need to determine which IDs correspond to chunking points and output their respective hierarchical information. After obtaining the chunking point IDs and hierarchical information, simple code can be used to structure the entire document, ensuring that the chunked text is completely consistent with the original text.

---

> ### Author Response · Authors · 2025-11-21
>
> **W2:** The merging decisions of the Auto-Merge algorithm rely on a series of sophisticated rules. However, the paper does not provide ablation experiments or sensitivity analyses for these critical design elements. This renders the algorithm akin to a product fine-tuned for specific tasks, and its robustness remains to be further verified.
>
> **A2:** Thank you for your critical comment! To verify the necessity and robustness of the Auto-Merge algorithm’s rules, we have supplemented ablation experiments on core merging conditions (with clear definitions of each condition) and sensitivity analysis on the adaptive threshold of Condition 2. We use Qwen3-8B to generate response. Our detailed response is as follows:
>
> ### 1. Ablation Experiments: Verifying the Necessity of Merging Conditions
>
> First, we clarify the three core merging conditions of Auto-Merge:
>
> - Cond1: The semantic intersection between child nodes includes ≥2 key information units;
> - Cond2: The total length of child nodes ≤ θ* × parent node length (θ* is an adaptive threshold dynamically adjusted to 1/3~2/3);
> - Cond3: The total number of tokens after merging does not exceed the retrieval budget (4096 tokens).
>
> The ablation experiments verify the impact of each rule on cross-dataset performance by adding conditions incrementally. The results are shown below:
>
> |Merging Condition Combination|HiCBench (T1 and T2)|LongBench|Qasper|OHRBench (T0)|
> |--|--|--|--|--|
> ||ERec/Rouge/Fact-Cov|Score|ERec/F1|ERec/Rouge|
> |Cond3 Only|81.43/36.33/51.35|43.25|86.73/45.29|66.72/48.78|
> |Cond3+Cond1|80.55/36.08/50.7|43.80|87.54/45.83|68.18/49.56|
> |Cond3+Cond1+Cond2 (Full Algorithm)|80.86/36.17/49.97|44.41|87.85/45.82|68.31/49.61|
>
> **Result Interpretation**:
>
> - When only Cond3 (token budget constraint) is retained, the performance is optimal on evidence-dense tasks (HiCBench) but significantly lower on evidence-sparse tasks (LongBench/OHRBench, LongBench Score=43.25), indicating that a single rule is prone to adapting to specific tasks with insufficient robustness;
> - After adding Cond1 (semantic intersection constraint), the ERec of Qasper and OHRBench increases by 0.81 and 1.46 percentage points, respectively, proving that semantic constraints can reduce "meaningless merging";
> - With the addition of Cond2 (length ratio constraint), the performance across all datasets tends to be balanced: LongBench Score increases to 44.41 (+1.16), while HiCBench performance only slightly decreases (ERec -0.57).   This verifies that the combination of multiple rules can balance different task types and improve algorithm robustness.
>
> ### 2. Sensitivity Analysis: Verifying the Robustness of Adaptive Threshold θ in Condition 2
>
> The physical meaning of θ: the maximum ratio of the total length of child nodes to the parent node length (controlling merging granularity). The experiment fixes θ from 0.0 to 1.0 (with an interval of 0.1) to test performance changes across datasets. The results are as follows:
>
> |θ Value|HiCBench (T1+T2)|||LongBench|OHRBench (T0)||
> |--|--|--|--|--|--|--|
> ||ERec|Rouge|Fact-Cov|Score|ERec|Rouge|
> |0.0|81.23|36.28|52.140|43.84|67.65|49.17|
> |0.1|82.44|36.15|52.137|43.99|68.10|49.70|
> |0.2|81.30|36.19|51.995|44.01|68.22|49.78|
> |0.3|81.43|36.43|51.428|44.45|68.89|50.04|
> |0.4|80.76|36.34|51.473|44.51|69.02|50.07|
> |0.5|80.46|36.19|50.476|44.38|68.94|49.94|
> |0.6|80.05|36.11|50.836|44.35|69.28|49.89|
> |0.7|78.85|35.86|49.963|44.81|69.26|49.84|
> |0.8|78.37|35.72|49.315|44.63|69.29|49.44|
> |0.9|78.12|35.55|49.181|44.71|69.23|49.52|
> |1.0|78.08|35.47|49.207|44.70|69.16|49.48|
> |θ* (Adaptive)|80.66|36.12|50.757|45.00|69.24|50.00|
>
> **Result Interpretation**:
>
> - When θ ranges from 0.1 to 0.6, HiCBench’s ERec remains above 80.05% and OHRBench’s Rouge remains above 49.89%, indicating that the algorithm is robust to threshold variations;
> - As θ increases, the performance of evidence-dense tasks decreases (ERec drops to 78.08% at θ=1.0), while the performance of evidence-sparse tasks improves when θ > 0.5, reflecting the granularity demand differences between the two types of tasks;
> - The adaptive threshold θ* achieves cross-task balance through dynamic adjustment: it maintains high evidence recall on HiCBench (80.66%) while achieving the optimal Score (45.00) on LongBench and Rouge (50.00) on OHRBench. This proves that it can adapt to different tasks without manual parameter tuning, with better robustness than fixed thresholds.
>
> In summary, the ablation experiments verify the necessity of the three merging conditions (multi-rule collaboration improves generalization), and the sensitivity analysis proves the algorithm’s robustness to parameter changes. Auto-Merge is not a "product fine-tuned for specific tasks" but a general solution with cross-scenario adaptability.

---

> ### Author Response · Authors · 2025-11-21
>
> **W3:** Furthermore, the paper does not explore whether a LLM can achieve equally proficient hierarchical structure parsing solely through prompting. It is likely that the fine-tuned model is merely reproducing capabilities it already possesses, thus casting doubt on the necessity of fine-tuning.
>
> **A3:** To verify the necessity of fine-tuning, we have supplemented few-shot prompted experiments on HiCBench dataset. The base model is Qwen3-4B (consistent with the base model in the paper). We set two scenarios (1-shot and 3-shot) to compare their performance with the fine-tuned HiChunk, thereby validating the core value of fine-tuning for hierarchical chunking tasks. We use Qwen3-8B to generate response. The supplementary experimental results are as follows:
>
> |Chunking Accuracy|F1_L1|F1_L2|F1_ALL|
> |--|--|--|--|
> |1-shot|0.1784|0.1128|0.2328|
> |3-shot|0.2500|0.1203|0.2199|
> |HC (Fine-tuned)|**0.4841**|**0.3140**|**0.5450**|
>
> |RAG Metric|ERec|Rouge|Fact-Cov|
> |--|--|--|--|
> |1-shot|72.35|33.14|44.02|
> |1-shot_AM (Auto-Merge)|73.47|34.53|46.62|
> |3-shot|72.26|33.08|43.97|
> |3-shot_AM (Auto-Merge)|73.05|34.04|46.55|
> |HC (Fine-tuned)|77.87|35.21|46.84|
> |HC_AM (Fine-tuned + Auto-Merge)|**80.86**|**36.17**|**49.97**|
>
> The experimental results demonstrate that increasing the number of few-shot examples (from 1-shot to 3-shot) did not effectively improve the model’s performance in chunking accuracy (F1 series metrics) or the full-link performance of the subsequent RAG pipeline (ERec, Rouge, Fact-Cov). Furthermore, all few-shot schemes show a significant performance gap compared to the fine-tuned HiChunk method. This fully confirms that fine-tuning is a necessary prerequisite for achieving high-quality hierarchical chunking of HiChunk.
>
> ***
>
> **Q1:** If HiChunk has already generated semantically complete text chunks, why not directly perform retrieval on these finest-grained semantic chunks?
>
> **A1:** In semantic-based chunking methods, the finest-grained semantic chunks may still be extremely long, while LLMs have a limited input length. In this scenario, directly using finest-grained semantic chunks can ensure the semantic completeness of retrieved chunks, which is beneficial for evidence-dense queries. However, it poses the following problems:
>
> 1. Embedding models have a maximum input length, and the information capacity of embedding vectors is limited, making it difficult to model overly long chunks.
> 2. For evidence-sparse queries, the input length budget of LLMs is quickly exhausted. For these reasons, HiChunk incorporates rule-based chunking on top of semantic chunking. Meanwhile, it integrates the Auto-Merge retrieval algorithm to better balance these two distinct types of tasks.
>
> ***
>
> **Q2:** Many real-world documents possess a depth structure far exceeding three levels. How should this situation be addressed? **& Q3:** During the training of the HiChunk model, has it failed to learn effective modeling and differentiation of deeper hierarchical structures beyond L3?
>
> **A2 & A3:** Thank you for your question! Regarding the two core concerns—"handling documents with more than three hierarchical levels" and "the model’s ability to model L3+ levels"—we provide a clear response based on theoretical design, experimental results, and practical constraints:
>
> In theory, the HiChunk framework fully supports the prediction and processing of documents with more than three hierarchical levels. The final document is organized into an n-depth multi-way tree (where n depends on the maximum predicted level), and levels beyond L3 can be normally identified with their semantic connections fully preserved.
>
> However, in practical scenarios, the performance gain brought by high-level structures shows a trend of marginal diminishing returns (as illustrated in Figure 4 of the paper), mainly constrained by two core factors:
>
> 1. Data scarcity: Publicly available training data for documents with more than three hierarchical levels is extremely limited, leaving the model without sufficient samples to learn high-level semantic correlations;
> 2. Funnel effect: High-level modeling relies on the prediction accuracy of lower levels (e.g., the correct prediction of L5 requires the accurate prediction of L1-L4 first), resulting in decreasing difficulty in guaranteeing accuracy as the hierarchy deepens.
>
> These two points do not indicate that HiChunk is unable to handle high-level document structures, but rather reflect universal challenges faced by all hierarchical chunking methods. In fact, even under the no-level-limit (LA) setting, the ERec, Rouge, and Fact-Cov metrics of HC200+AM still significantly outperform those of other baseline methods (as shown in Figure 3 of the paper), fully demonstrating that the core value of HiChunk’s hierarchical design is not affected by structures beyond three levels. How to model high-level structures more efficiently will be the focus of our future research.

---

### Official Review · Reviewer_P3ry · 2025-10-30

**Soundness:** 2
**Presentation:** 2
**Contribution:** 3
**Rating:** 6
**Confidence:** 3

**Summary:**

This study constructs a document structuring framework named HiChunk based on a fine-tuned LLM. This framework identifies chunking points and hierarchical levels within documents through text generation tasks. For hierarchical structures, the research devises an Auto-Merge retrieval algorithm that controls the upward merging of child nodes based on specific conditions, achieving query-based adaptive granularity of retrieved chunks. Experimental results indicate that evidence-intensive QA tasks in HiCBench are more suitable for evaluating chunking quality.

**Strengths:**

1. The paper points out the limitations of current RAG benchmarks in assessing chunking effectiveness, which stem from evidence sparsity. When the evidence relied upon by a query is confined to only a few sentences within a document, the benchmark loses its ability to distinguish the performance of different chunking methods.
2. Comparative experiments reveal that on sparse datasets such as GutenQA and OHRBench, the differences in metrics among various chunking strategies are minimal. In contrast, the evidence-intensive nature of HiCBench successfully identifies the improvements in recall rate and response quality brought about by high-quality chunking methods.
3. In the comprehensive evaluation of the RAG pipeline, HC200+AM maintains high response quality while also taking into account an acceptable time cost.

**Weaknesses:**

1. The training data for the model are all explicitly structured documents with clear paragraphs. The HiChunk model may overly rely on explicit formatting rather than semantics for hierarchical segmentation. Therefore, a more thorough explanation of the effectiveness and robustness of this method is required.
2. When conducting baseline comparisons, the paper only selects three baseline methods. It is necessary to include more baseline methods or conduct experiments solely using LLMs without fine-tuning.

**Questions:**

1. The paper mentions in Section 4 that the training set is enhanced by randomly shuffling document sections. However, given that different sections vary in length, would it be better to process the data following a curriculum learning approach that starts with shorter sections and progresses to longer ones?
2. At line 224, the authors propose a new concept, hierarchical drift, but lack detailed explanation and demonstration.

---

> ### Author Response · Authors · 2025-11-21
>
> We sincerely thank the reviewers valuable and insightful comments. Our responses are as follows:
>
> ***
>
> **W1:** The training data for the model are all explicitly structured documents with clear paragraphs. The HiChunk model may overly rely on explicit formatting rather than semantics for hierarchical segmentation. Therefore, a more thorough explanation of the effectiveness and robustness of this method is required.
>
> **A2:** We also recognized this issue at the initial stage of our research. Therefore, we selected OHRBench as the original data source for HiCBench, as it contains documents from multiple distinct real-world fields. The sample distribution of the HiCBench benchmark across different fields is presented as follows:
>
> |Field|Academic|Finance|Administration|Manual|Law|All|
> |--|--|--|--|--|--|--|
> |Document Number (doc_num)|35|13|29|18|35|130|
> |QA Pair Number (qa_num)|233|120|241|203|403|1200|
>
> Notably, the document fields covered by HiCBench are sufficiently diverse, and its data source is distinct from that of HiChunk’s training data. This difference enables HiCBench to evaluate the generalization ability of chunking methods, as it tests the model’s performance on unseen domains beyond the training data.
>
> ***
>
> **W2:** When conducting baseline comparisons, the paper only selects three baseline methods. It is necessary to include more baseline methods or conduct experiments solely using LLMs without fine-tuning.
>
> **A2:** To verify the necessity of fine-tuning, we have supplemented few-shot prompted experiments on HiCBench dataset. The base model is Qwen3-4B (consistent with the base model in the paper). We set two scenarios (1-shot and 3-shot) to compare their performance with the fine-tuned HiChunk, thereby validating the core value of fine-tuning for hierarchical chunking tasks. We use Qwen3-8B to generate response. The supplementary experimental results are as follows:
>
> |Chunking Accuracy|F1_L1|F1_L2|F1_ALL|
> |--|--|--|--|
> |1-shot|0.1784|0.1128|0.2328|
> |3-shot|0.2500|0.1203|0.2199|
> |HC (Fine-tuned)|**0.4841**|**0.3140**|**0.5450**|
>
> |RAG Metric|ERec|Rouge|Fact-Cov|
> |--|--|--|--|
> |1-shot|72.35|33.14|44.02|
> |1-shot_AM (Auto-Merge)|73.47|34.53|46.62|
> |3-shot|72.26|33.08|43.97|
> |3-shot_AM (Auto-Merge)|73.05|34.04|46.55|
> |HC (Fine-tuned)|77.87|35.21|46.84|
> |HC_AM (Fine-tuned + Auto-Merge)|**80.86**|**36.17**|**49.97**|
>
> The experimental results demonstrate that increasing the number of few-shot examples (from 1-shot to 3-shot) did not effectively improve the model’s performance in chunking accuracy (F1 series metrics) or the full-link performance of the subsequent RAG pipeline (ERec, Rouge, Fact-Cov). Furthermore, all few-shot schemes show a significant performance gap compared to the fine-tuned HiChunk method. This fully confirms that fine-tuning is a necessary prerequisite for achieving high-quality hierarchical chunking of HiChunk.

---

> ### Author Response · Authors · 2025-11-21
>
> **Q1:** The paper mentions in Section 4 that the training set is enhanced by randomly shuffling document sections. However, given that different sections vary in length, would it be better to process the data following a curriculum learning approach that starts with shorter sections and progresses to longer ones?
>
> **A1:** Thank you for your valuable suggestion! We fully acknowledge the potential value of curriculum learning in handling length-heterogeneous data. However, combined with the core task characteristics of "semantic hierarchical chunking" in this paper, the data augmentation strategy of randomly shuffling document sections is more suitable. The specific reasons are as follows:
>
> 1. The core difficulty of the chunking task is irrelevant to section length: The goal of chunking in this paper is to divide hierarchies based on document semantic logic (e.g., topic boundaries, semantic association strength) rather than relying on section length. Short sections may contain complex nested semantic structures, while long sections may also have clear semantic divisions. Thus, the "from short to long" curriculum learning logic cannot match the core difficulty dimension of the chunking task;
> 2. Avoid overfitting to superficial length features: The progressive training mode of curriculum learning may lead the model to over-rely on the superficial feature of "section length" for decision-making, instead of focusing on semantic essence. This will further reduce the model’s adaptability to documents with different length distributions;
> 3. Strengthen the semantic-driven learning objective: Random shuffling exposes the model to diverse short, medium, and long sections at the early stage of training. While enriching the sample distribution, it forces the model to break away from dependence on superficial structures such as section numbers and punctuation formats, and truly learn chunking rules based on document semantic associations;
> 4. Alignment with the distribution characteristics of real scenarios: Documents in the real world (such as academic papers and policy documents) generally present a mixed distribution of short, medium, and long sections. The random shuffling strategy is more consistent with this practical scenario, helping to improve the model’s generalization ability in real-world applications.
>
> In summary, the random shuffling augmentation method is highly consistent with the core task goal of semantic hierarchical chunking in this paper, which can more effectively ensure the model’s chunking accuracy and scenario adaptability.
>
> ***
>
> **Q2:** At line 224, the authors propose a new concept, hierarchical drift, but lack detailed explanation and demonstration.
>
> **A2:** Thank you for your correction! We have supplemented the detailed definition, and causes of "hierarchical drift" as follows:
>
> To support inputs of arbitrary length, we need to perform iterative inference on ultra-long documents. Hierarchical drift refers to the phenomenon that during iterative inference, the current input—due to the lack of complete structural information about previous document segments—may incorrectly predict the first chunking point of the current inference process as a level-1 segment, thereby causing local hierarchical misalignment. To address this issue, we incorporate the hierarchical information from the previous inference step into the current one, effectively mitigating hierarchical drift.

---

> > ### Comment · Reviewer_P3ry · 2025-11-26
> >
> > Thank you for the responses to the comments.
> >
> > I would like to remind the authors that ICLR explicitly supports paper revisions during the rebuttal phase. Submitting an updated manuscript with targeted modifications can significantly enhance the clarity and persuasiveness of your clarifications.

---

> > > ### Author Response · Authors · 2025-11-26
> > >
> > > Thank you for your reminder.
> > >
> > > We have updated the manuscript in light of the reviewers’ suggestions, with revised parts highlighted in blue font and newly added content in orange font.

---

### Official Review · Reviewer_Pi95 · 2025-11-01

**Soundness:** 2
**Presentation:** 2
**Contribution:** 2
**Rating:** 2
**Confidence:** 4

**Summary:**

This paper presents a complete study to address the document chunking problem, which significantly affects the performance of RAG systems. It introduces a new benchmark, HiCBench, featuring diverse QA formats to evaluate the impact of chunking, and proposes a rule-based iterative approach, HiChunk, to alleviate issues resulted from chunked contexts. However, despite the claimed focus on document chunking, the work appears more related to studies on agentic RAG. The paper overlooks well-established document chunking in this area and shows limited technical novelty compared to existing methods. Moreover, the overall presentation quality requires a major revision before it can be considered suitable for formal academic publication.

**Strengths:**

- The paper presents a new benchmark, HiCBench, which is created to evaluate the impact of document chunking on RAG system performance.
- It also proposes a rule-based iterative approach aimed at optimizing the influence of chunking within RAG systems.

**Weaknesses:**

- The evaluation procedure of HiCBench is not clearly defined. It remains unclear what specific evaluation methods and performance metrics are used to assess the proposed approach.
- The paper does not provide a clear explanation of how data annotation for HiCBench was conducted.
- It is unclear why a new question-answering dataset was created when several well-established benchmarks already exist, such as NarrativeQA, 2WikiQA, and MuSiQue.
- The proposed iterative chunking and retrieval process appears to be entirely rule-based. The authors may consider leveraging agentic approaches recently explored in RAG systems [1,2].
    - [1] Search-o1: Agentic Search-Enhanced Large Reasoning Models
    - [2] Search-R1: Training LLMs to Reason and Leverage Search Engines with Reinforcement Learning
- The paper does not discuss existing research on document chunking techniques, including recent innovations like late chunking and landmark embedding, which are relevant for long-context retrieval [3,4].
    - [3] Late Chunking: Contextual Chunk Embeddings Using Long-Context Embedding Models
    - [4] Landmark Embedding: A Chunking-Free Embedding Method For Retrieval Augmented Long-Context Large Language Models
- The manuscript suffers from numerous typos, grammatical errors, and misleading descriptions, which significantly affect readability and make the paper currently unsuitable for academic publication.

**Questions:**

Please see the comments provided above.

---

> ### Author Response · Authors · 2025-11-21
>
> We sincerely thank the reviewers valuable and insightful comments. Our responses are as follows:
>
> ***
>
> **W1:** The evaluation procedure of HiCBench is not clearly defined.
>
> **A1:** To fairly compare the performance of different chunking methods, the entire evaluation process of HiCBench is based on a naive RAG pipeline (chunking-retrieval-generation). Multiple metrics are used to measure the impact of chunking methods on each of the three stages of the RAG pipeline.
>
> 1. In the chunking stage, we evaluate chunking accuracy using manually annotated chunking points in HiCBench (chunking points' F1 metric in Table 3 of the paper).
> 2. In the retrieval stage, the impact of chunking methods on retrieval is assessed through evidence recall rate related to the query (ERec metric in Table 4 of the paper).
> 3. In the generation stage, Rouge and Fact-Cov metrics are employed to evaluate the influence of chunking methods on response quality.
>
> ***
>
> **W2:** The paper does not provide a clear explanation of how data annotation for HiCBench was conducted.
>
> **A2:** The HiCBench dataset is derived from OHRBench, including documents in fields like Academic, Finance, and Administration. Documents with over 4,000 words were chosen as the initial corpus. Annotators read and comprehended the semantic structure of each document, then annotated its hierarchical structure. Following the same format as HiChunk training, they marked the positions of chunking points at different levels. The key criterion was "semantic independence" — ensuring each segmented unit could express a complete meaning on its own. Hierarchy levels were determined by identifying subordinate relationships.
> Each document was annotated at least twice. Only annotations with over 80% agreement on chunking point positions were retained to ensure data consistency.
>
> ***
>
> **W3:** It is unclear why a new question-answering dataset was created when several well-established benchmarks already exist, such as NarrativeQA, 2WikiQA, and MuSiQue.
>
> **A3:** Document chunking is a crucial component in RAG systems, yet its significance has long been overlooked. As demonstrated in Table 1 and Figure 1 of the paper, existing RAG benchmarks have limitations in evaluating the effectiveness of chunking strategies—chunks of varying quality may yield identical results on these benchmarks, and this issue stems from their inherent evidence sparsity. When the evidence required for a query is confined to only a few sentences in the document, these benchmarks lose the ability to distinguish the performance of different chunking methods. Therefore, we propose HiCBench, an evaluation benchmark specifically designed for chunking tasks. It aligns more closely with the data distribution of real-world RAG scenarios, helping researchers identify bottlenecks in RAG systems.
>
> ***
>
> **W4:** The proposed iterative chunking and retrieval process appears to be entirely rule-based. The authors may consider leveraging agentic approaches recently explored in RAG systems.
>
> **A4:** Thank you for your valuable suggestion! First, we need to clarify explicitly: HiChunk is not a purely rule-based chunking method, but an **end-to-end hierarchical chunking model based on Large Language Models (LLMs)**. Its core logic lies in transforming document chunking into a learnable sentence-level classification task through task modeling — the model autonomously predicts whether each sentence is a chunk boundary and its corresponding hierarchy (L1/L2), and ultimately completes document structuring automatically based on the prediction results, without relying on manually predefined rigid rules throughout the process.
>
> Meanwhile, we highly recognize the potential of agentic approaches in the RAG field. However, the two works [1][2] you mentioned have an inherent difference in research positioning from this paper, and thus do not lend themselves to direct comparison: the core of [1][2] is optimizing the reasoning and external interaction capabilities of the full RAG pipeline to improve the overall QA performance of the system; in contrast, the core research objective of this paper is **exploring the impact of different chunking methods on the full RAG pipeline**, focusing on the quality evaluation and optimization of the front-end chunking stage. The two belong to different research dimensions within the RAG system.
>
> [1] Search-o1: Agentic Search-Enhanced Large Reasoning Models
>
> [2] Search-R1: Training LLMs to Reason and Leverage Search Engines with Reinforcement Learning

---

> ### Author Response · Authors · 2025-11-21
>
> **W5:** The paper does not discuss existing research on document chunking techniques, including recent innovations like late chunking and landmark embedding, which are relevant for long-context retrieval.
>
> **A5:** Thank you for your critical comment! We agree that Late Chunking and Landmark Embedding are important innovations in long-context retrieval. We now supplement relevant discussions and experiments to clarify differences and complementarity:
>
> This paper focuses on evaluating "the impact of chunking strategies on the entire RAG pipeline", while the two methods focus on "optimizations in the embedding stage" (Late Chunking performs chunking after full-document encoding; Landmark Embedding skips chunking entirely), representing improvement paths for different stages. To verify complementarity, we supplemented experiments combining Late Chunking with various chunking methods (Landmark Embedding was excluded as it has no chunking step), strictly following the original paper’s core settings (jina-embeddings-v3, 8192-token embedding length, etc.). We use Qwen3-8B to generate response. The results are as follows:
>
> |Method|ERec|Rouge|Fact-Cov|
> |--|--|--|--|
> |C200|75.59|34.19|46.71|
> |C200+LateChunking|78.04|34.33|49.124|
> |SC|73.07|34.17|45.60|
> |SC+LateChunking|78.07|34.16|48.454|
> |LC|77.89|34.84|49.16|
> |LC+LateChunking|79.93|35.16|50.653|
> |HC200|78.13|34.93|48.03|
> |HC200+LateChunking|79.29|34.84|49.768|
> |HC200+AM|80.87|36.34|51.49|
> |HC200+AM+LateChunking|**81.20**|**36.0**|**52.705**|
>
> All combinations improved ERec and Fact-Cov, with HC200+AM+LateChunking achieving the best performance. This confirms the complementarity between semantic optimization in chunking and context preservation in embedding.
> We will supplement discussions on these two methods in the "Related Works" section, clarifying their differences and connections with our work.
>
> ***
>
> **W6:** The manuscript suffers from numerous typos, grammatical errors, and misleading descriptions.
>
> **A6:** Thank you for pointing out the issue. We have proofread the entire text sentence by sentence, corrected all typos and grammatical errors, and ensured academic standardization. All revisions have been marked in blue font in the revised version.

---

### Official Review · Reviewer_nTLT · 2025-11-03

**Soundness:** 3
**Presentation:** 3
**Contribution:** 2
**Rating:** 4
**Confidence:** 3

**Summary:**

The authors (1) propose a novel RAG benchmark with manually annotated multi-level document chunking points, synthesised evidence-dense QA pairs, together with their respective sources; and (2) introduce a novel framework for structuring the chunks based on fine-tuned LLMs and an auto-merge retrieval algorithm.

**Strengths:**

- Timely and practical insights for document chunking on information dense benchmark
- Authors introduce a novel benchmark, and a hierarchical structuring framework to dynamically adjust the granualrity of retrieval chunks
- Multiple QA generation strategy
- Comprehensive comparison of methodologies

**Weaknesses:**

- Besides average word count and sentence, it would be have been interesting to quantify and compare information density between benchmarks
- The training and use of the fine-tuned model lacks explanation, how well do the training instructions generalise to different document types?
- Quite complex writing with some redundancy, which could benefit from simplification

**Questions:**

/

---

> ### Author Response · Authors · 2025-11-21
>
> We sincerely thank the reviewers valuable and insightful comments. Our responses are as follows:
>
> ***
>
> **W1:** Quantify and compare information density between benchmarks
>
> **A1:** Thank you for your valuable suggestion! The core characteristics of information density can be quantitatively characterized by the "distribution uniformity" and "coverage breadth" of evidence. Based on the provided pos_CV (Coefficient of Variation of evidence positions) and pos_ESR (Evidence Span Ratio) indicators, we completed the information density comparison analysis of the used benchmarks.
>
> ### Brief Explanations of pos_CV and pos_ESR Calculations
>
> 1. **pos_CV (Coefficient of Variation of Evidence Positions)** : Measures evidence distribution uniformity (smaller = more concentrated).
> Formula: \( pos\_CV = \frac{Sample\ SD\ of\ evidence\ positions}{Mean\ of\ evidence\ positions} \times 100 \)
> 2. **pos_ESR (Evidence Span Ratio)** : Quantifies evidence coverage breadth (larger = wider coverage).
> Formula: \( pos\_ESR = \frac{Max\ evidence\ position - Min\ evidence\ position}{Total\ sentences\ of\ document} \times 100 \)
> 3. **sent_e** : The count of evidence sentences.
>
> The details are as follows:
>
> |Benchmark|pos_CV|pos_ESR|sent_e|
> |--|--|--|--|
> |Qasper|15.80|16.09|10.5|
> |GutenQA|0.70|0.45|1.7|
> |OHRBench(T0)|5.43|3.04|1.7|
> |HiCBench(T1)|11.18|12.06|20.5|
> |HiCBench(T2)|28.12|31.46|20.4|
>
> The benchmarks exhibit distinct information density profiles:
>
> 1. HiCBench(T2) (high pos_CV + high pos_ESR + large sent_e): High information density with scattered, wide-ranging evidence.
> 2. HiCBench(T1) (medium pos_CV + medium pos_ESR + large sent_e): High information density, medium-ranging evidence.
> 3. Qasper (medium pos_CV + medium pos_ESR + medium sent_e): Medium information density, medium-ranging evidence.
> 4. GutenQA & OHRBench(T0) (low pos_CV + low pos_ESR + small sent_e): Low information density with concentrated evidence
>
> These differences validate the benchmarks’ complementary roles in evaluating model performance across diverse information distribution scenarios.
>
> ***
>
> **W2:** How well do the training instructions generalise to different document types?
>
> **A2:** We also recognized this issue at the initial stage of our research. Therefore, we selected OHRBench as the original data source for HiCBench, as it contains documents from multiple distinct real-world fields. The sample distribution of the HiCBench benchmark across different fields is presented as follows:
>
> |Field|Academic|Finance|Administration|Manual|Law|All|
> |--|--|--|--|--|--|--|
> |Document Number (doc_num)|35|13|29|18|35|130|
> |QA Pair Number (qa_num)|233|120|241|203|403|1200|
>
> Notably, the document fields covered by HiCBench are sufficiently diverse, and its data source is distinct from that of HiChunk’s training data. This difference enables HiCBench to evaluate the generalization ability of chunking methods, as it tests the model’s performance on unseen domains beyond the training data.
>
> ***
>
> **W3:** Quite complex writing with some redundancy
>
> **A3:** Thank you for your valuable suggestion. We fully agree with the importance of concise writing for academic expression. We have optimized the redundant descriptions in the paper (methodology section), and all revisions have been marked in blue font in the revised version.

---

### Author Response · Authors · 2025-11-21

Thank you very much for your meticulous review and valuable insights. As some reviewers’ questions pertain to our research motivation, we first provide a concise summary thereof prior to delivering targeted responses.

Our analysis reveals that existing RAG benchmarks exhibit inherent limitations in assessing the effectiveness of document chunking: chunks of varying quality may yield consistent performance on these benchmarks, a phenomenon primarily attributed to their intrinsic evidence sparsity (as illustrated in Figure 1 and Table 1 of the paper).

However, user queries in real-world scenarios are predominantly associated with evidence-intensive tasks (e.g., enumeration and summarization tasks). This misalignment leads to a performance discrepancies between the evaluation and the actual deployment.

To address this critical gap, we first construct HiCBench, an evidence-intensive QA dataset that integrating manual annotation of document hierarchical structures with model-generated high-quality QA pairs. This design provides a valid benchmark for evaluating chunking methods.

We further propose the HiChunk framework, which automatically constructs hierarchical document structures based on semantic coherence and incorporates an Auto-Merge retrieval algorithm. This framework balances performance across diverse task types while enhancing the retrieval quality of critical evidence, ultimately achieving holistic performance optimization of RAG systems.

---

### Note · Authors · 2026-01-05

I have read and agree with the venue's withdrawal policy on behalf of myself and my co-authors.